# The burden of non-communicable disease risk factors in a low-income population: Findings from a cross-sectional study highlighting the prevalence of obesity, hypertension, and metabolic disorders in the south of Quito, Ecuador

Sergio Morales-Garzón[1]*, Juan Vasconez[2], Jessica Pinto Delgado[3], Francisco Barrera-Guarderas[4], Elisa Chilet-Rosell[1,5], Marta Puig-García[1,5], Andrés Peralta[3,4], María Fernanda Rivadeneira Guerrero[3,4], Ana Lucía Torres-Castillo[3], Lucy Anne Parker[1,5]

1 Department of Public Health, History of Science and Gynaecology Department, Miguel Hernández University, Alicante, Spain, 2 Faculty of Nursing, Pontificia Universidad Católica del Ecuador, Quito, Ecuador, 3 Institute of Public Health, Pontificia Universidad Católica del Ecuador (PUCE), Quito, Ecuador, 4 Faculty of Medicine, Pontificia Universidad Católica del Ecuador, Quito, Ecuador, 5 CIBER in Epidemiology and Public Health (CIBERESP), Madrid, Spain

☯ All the authors contributed equally to this work.

* s.morales@umh.es

## Abstract

### Objectives

We describe the prevalence of Non-Communicable Disease (NCD) risk factors in a low-income health district in the South of Quito, Ecuador.

### Study design

We conducted a cross-sectional study between November 2020 and October 2021

### Methods

We used multi-stage cluster sampling to select 656 of población total adult residents of 17D06 health district, enabling a prevalence estimation with at least ±5.73% absolute precision. We collected socio-demographic information and behavioural risk factors using the expanded WHO STEPwise approach to NCD surveillance. We measured height, weight, and blood pressure, and collected blood samples to assess glucose levels, lipid profiles, and renal function. We estimated the prevalence of behavioural and metabolic NCD risk factors by sex and age groups (18–44, 45–69, and >70).

**Data availability statement:** The raw data, codebook, and survey tool used for this study are available in Zenodo (DOI: 10.5281/zenodo.13889954).

**Funding:** This project has received funding from the European Research Council (ERC) under the European Union's Horizon 2020 research and innovation programme (Grant agreement No. 804761). The funders had no role in study design, data collection and analysis, decision to publish, or preparation of the manuscript.

**Competing interests:** The authors have declared that no competing interests exist.

## Results

One-third of participants were obese (33.2% overall, 148 women, 38,7%, and 41 men, 22%), and more than half had a raised waist circumference (56.8%, N = 322). Hypertension affected 26.9% of participants (63 men, 33% and 90 women, 24%). Hyperglycaemia affected 7.9% (N = 45) of participants and increased with age and peaked at 22% among women over 70. More than half of the participants presented hypercholesterolemia (317 individuals, 56.2%). Low consumption of fruit and vegetables, high salt consumption and high sugar consumption were common in all population groups (88.4%, N = 580, 33.2%. N = 218 and 72.4%, N = 475, respectively).

## Conclusions

The critical prevalence of NCD risk factors in this low-income urban district of Quito, alongside similar trends observed in other settings across Latin America, underscores the need for ecological public health strategies to create healthy environments and promote healthier behaviours. Gender-sensitive approaches may be useful to address differences between sexes.

## Introduction

Health disparities are one of the major determinants of disability and disease [1,2]. According to the World Health Organization (WHO) and other public health agencies [3–5], these disparities not only affect the rise of chronic health conditions or mental health problems, but also interact with environmental factors further exacerbating the situation of inequity [6,7]. For example, in an obesogenic environment, low socio-economic status and problems with accessibility may make unhealthy behaviours into more affordable options. Recognising the ubiquitous existence of such environments and how they contribute to the escalation of behavioural risk factors associated with Non-Communicable Diseases (NCDs), indicates that prevention and control strategies should be oriented towards ecological strategies rather than individual strategies [8,9].

Low- and middle-income countries were recently situated as principal territories where NCDs are more prevalent [4,10,11]. According to the data, 70% of adult deaths in 2019 were caused by NCDs [12], which are mainly cardiovascular diseases, diabetes, cancer, obesity and chronic respiratory diseases. In Ecuador, where this research carried out, 78% of deaths are caused by NCDs, with a significant gender gap: while 7 out of 10 deaths in men are caused by NCDs, women account for 9 out of 10 deaths in women [12].

In the last four decades, Latin America has experienced a dietary transition that has altered the foundation of nutrition [13]. The evolution of consumption patterns, the proliferation of unhealthy habits, and the double burden of malnutrition, understood as the coexistence of undernutrition and overweight/obesity within the same

population [14], describe a scenario where environmental factors of dietary patterns, such as accessibility, availability, and healthiness, define a population with alarming epidemiological rates.

The WHO has developed the standardised STEPwise approach [15] for surveillance of NCD risk factors, to promote comparability of data on key NCD risk factors between countries or regions, and over time. In STEPS Ecuador 2018, with 4.638 nationally representative participants aged between 18–69, the results showed a prevalence of hypertension, diabetes, and hypercholesterolemia of 20%, 7% and 34% respectively [16]. Furthermore, it uncovered an alarming problem with dietary habits. Regarding the consumption of fruits and vegetables, only 5% of the population reported consuming five or more portions per day, and 76% of the population reported adding salt regularly to their food. The report also indicated a high consumption of sugary drinks and that around 1 in 4 people reported insufficient physical activity [16].

The current study stems from a local health district level STEPS survey carried out as part of the CEAD project [17], which aims to provide rigorous epidemiologic data on NCD risk and morbidity in both urban and rural low-income districts of Ecuador and explore contextualisation of global recommendations for control, specifically diabetes, in low resource settings. Given that evidence shows NCDs and their risk factors are highly influenced by environmental aspects, we sought to gather local level data in a low-income urban district of Quito. Here, we describe the prevalence of NCD risk factors stratified by age and sex in a low-income district located in the south of Quito.

## Methods

### Study design

We conducted a cross-sectional study with a sample of 656 residents, over 18 years of age, from the 17D06 health district in the South of Quito, Ecuador. We recruited participants from November 2020 to October 2021 using a multi-stage cluster sampling design and implemented the expanded WHO STEPwise approach to surveillance of non-communicable diseases survey [15].

### Setting

This study was carried out in Quito, the capital of Ecuador, located in the Andes at an altitude of 2,850 metres. The city has a tempered tropical mountain climate, with a dry season from June to September and rainy from October to May. The city, which is expected to be among the fastest-growing cities (1–3% annually) over the next decade [18], faces environmental challenges resulting from urban expansion, such as the deterioration of air quality driven by the increasing use of private transport and the consumption of high-sulphur-content fuels [19] which directly impacts people's health [20], as well as the vulnerability of its ecosystems [21]. Socially, Quito is complex and diverse city. Despite government efforts to reverse inequality, its population of over two million includes precarious neighbourhoods where service shortages, infrastructure deficits, and poverty accentuate the segregation between residents of the north and the south [22]. The health district 17D06 is in the south of the city, home to over one million inhabitants and characterised by limit access to public services, such as health or education, driving to an economy based on informal employment and social inequality [23,24] (Fig 1).

### Participants

The inclusion criteria for participating in the survey were being aged over 18 and residing in the 17D06 health district. We initially proposed a sample size of 720 participants, assuming a diabetes prevalence of no more than 10%—in line with the principal objective of the CEAD project—to allow estimation with an absolute precision of ±3% at a 95% confidence level. A design effect of 1.5 was applied, as recommended by the WHO STEPS guidelines for complex survey designs in the absence of locally available alternatives. The sample size was further increased to account for up to 20% non-response. The final achieved sample size was 656. This sample allows estimation of the prevalence of any dichotomous variable in the overall population with an absolute precision of at least ±5.73%, assuming the same design effect of

1.5.The multistage cluster sampling approach was based on the land registry and involved three stages. In the first stage, we randomly selected 60 of the 1024 Census Sections (CS) in the health district with a probability proportional to population size. In the second stage, we randomly selected 20 GPS points from within the urbanised areas of each CS (12 GPS points at intended recruitment spots, and 8 potential substitute GPS points to use as necessary, 1200 GPS points in total). In the third stage, we visited the building closest to each GPS point randomly selected one household in multi-household buildings and randomly selected 1 person aged over 18 per household to be invited to participate in the survey. A total of 996 individuals were invited to participate, a total of 319 individuals (32%) refused to participate in the study. This high refusal rate was partly due to the pandemic situation, where fear and recommendations for isolation made people more reticent to invite the survey team into their homes. Of those who agreed to participate 656 (65,9%) completed the survey questionnaire and 567 (85.4%) also completed biological and physical measurements (Fig 2).

## Recruitment and procedures

Data collection took place from November 2020 to October 2021. The survey team consisted of 4 interviewers, a team supervisor, and a survey coordinator. Each morning, the coordinator and team supervisor assigned the interviewers a list of GPS points to visit. Data were collected using tablets with the Kobo Toolbox application. This survey included a

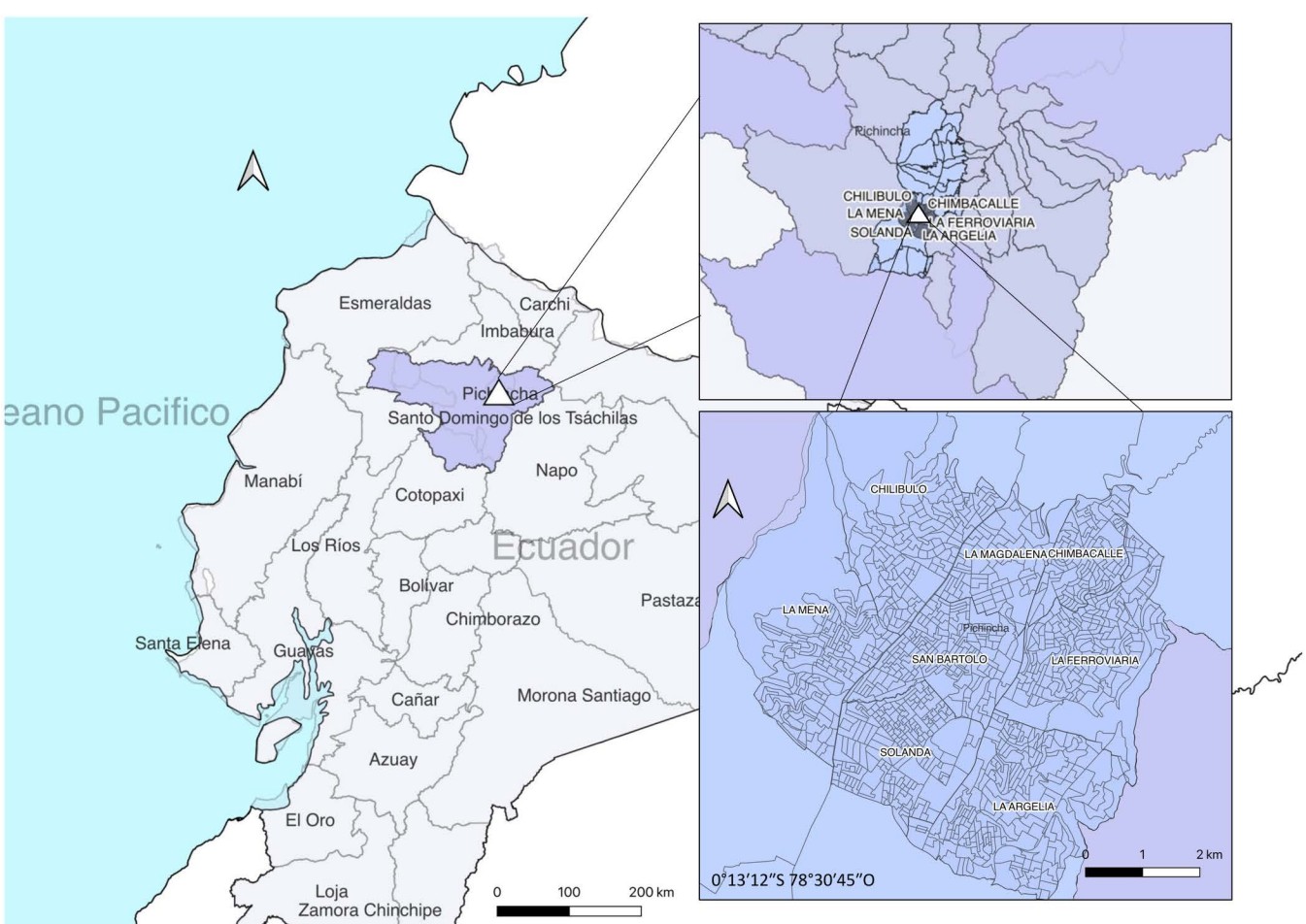

**Fig 1. Health district 17D06 located in Quito, Pichincha province, Ecuador.**

questionnaire as well as physical and biological measurements. The team made household visits to fill the questionnaire and scheduled a second home visit for physical measurements and blood sampling. Participants were instructed to fast for at least 8 hours for the biological measurements. Before starting the questionnaire, people gave informed consent after learning about the objectives of the project and what their participation consisted of (including risks and benefits). Participants' responses were recorded on digital tablets (Samsung Galaxy Tab AT290) with purpose designed forms using the programme KoboCollect (version 2.4).

### Questionnaire

We defined our questionnaire according to the definitions and recommendations of the STEPwise tool user manual [15] with cultural adaptations. For example, we modified the education-related questions to align with the Ecuadorian system, incorporated relevant ethnic groups, and added culturally appropriate visual prompts to the dietary section of the questionnaire, such as using familiar processed food brands and locally recognized fruits. The questionnaire included 6 sections: i) sociodemographic characteristics (educational attainment, ethnicity, marital status, employment status and household income), ii) tobacco consumption, iii) alcohol consumption, iv) dietary components, v) physical activity, and vi) medical history and health care received. The Spanish version of the questionnaire used is available in supplementary material. (S2 File)

### Physical and biological measurements

In the second home visit, which was previously agreed with each participant, the survey team returned with a mobile laboratory technician from a nationally accredited clinical laboratory (SYNLAB Solutions in Diagnostics, Quito, Ecuador) to conduct physical and biological measurements. We reminded participants that fasting was required for the proper functioning of the tests. We asked them whether they were receiving insulin or antidiabetic treatment, had been diagnosed with haemophilia, were undergoing chemotherapy or suspected they might be pregnant. We provided participants with urine beakers to test for glucosuria prior to the start of the test and analysed urine glucose concentration using test strips. The oral glucose tolerance test was offered only to individuals who were not undergoing diabetes treatment or chemotherapy, and did not have haemophilia, were not pregnancy, and did not present urine glucose levels consistent with diabetes.

We measured weight, height, waist circumference and blood pressure (measured with Riester mod. Ri-Champion nº 08001946−83 with appropriate armband size). The laboratory technician drew a first blood sample to test fasting blood glucose, total cholesterol, high density lipoprotein cholesterol (HDL), low density lipoprotein cholesterol (LDL), triglycerides and creatinine. For participants who underwent the Oral Glucose Tolerance Test (Glutest, Quimical EC), a second blood sample was collected two hours after the first, following the ingestion of 75 g of glucose.

We sought to collect the physical and biological measurements of up to 12 people per morning. Whole blood samples were centrifuged at 1800g revolutions per minute for 10 minutes within 30 minutes of extraction and serum samples were stored in an icebox at 2–8°C until transportation to the laboratory. Serum samples were processed analysed within 2 hours of arrival at the laboratory using standard protocols [25].

### Behavioural and metabolic risk variables

For fruit and vegetable consumption, we assessed whether individuals consumed more or less than 5 pieces per day, following WHO recommendations. Similarly, we analysed physical activity according to WHO recommendations, which suggest at least 150 minutes of moderate intensity aerobic physical activity per week. Questions considered any activity, including leisure time activities, transport to and from places, or activities as part of a person's job.

Regarding physical measurements, we calculated the Body Mass Index (BMI) defining categories following the standard model (overweight as a BMI ≥ 25 kg/m2 and < 30 kg/m2, and obesity as a BMI ≥ 30 kg/m2) [26]. Waist circumference (WC) was defined as raised if it was higher than 94 cm for men and 90 cm for women [27]. We defined hypertension

as having a systolic blood pressure (SBP) ≥ 140 mmHg and diastolic blood pressure (DBP) ≥ 90 mmHg or those who reported receiving pharmacological treatment for raised blood pressure.

After analysis of the samples in the laboratory, we obtained the data for glycaemic analysis, classifying the results into normoglycemia (PBG < 140 mg/dL or FBG > 70- < 99 mg/dL), Impaired glucose tolerance (PBG > 140- < 199 mg/dL or FBG > 100- < 126 mg/dL) and Hyperglycaemia (PBG > 200 or FBG > 126 mg/dL). For the posterior analysis, we considered individuals as hyperglycaemic if their laboratory results presented diabetes compatible values or if they reported receiving pharmacological treatment for glycaemic control. Total cholesterol was defined as raised when levels were over 190 mg/dL, HDL was considered lower when its level was less than or equal to 50 mg/dL in women and less than or equal to 40 mg/dL in men. LDL was considered raised when its level was equal to or over 115 mg/dL and Triglycerides were defined as raised when its level was over 150 mg/dL. We defined normal creatinine as levels between 0.7 mg/dL and 1.3 mg/dL in men and 0.6 mg/dL to 1.1 mg/dL in women.

Finally, we estimated individual cardiovascular risk using the WHO tables of 2019 for Andean Latin America [28] which incorporate age, sex, SBP, smoking, cholesterol, and diabetes status, aiming to provide evidence of the 10-year odds of developing cardiovascular diseases as an indicator of population health. Those who had inconclusive results of total cholesterol, diabetes or blood pressure were excluded from the CVD analysis. A table with definitions and cut-off points is available in supplementary Table S1 (S1 File).

### Data analysis

We performed the data cleaning using Microsoft Excel (Microsoft Corporation, 2018), and the analysis using Stata (Stata Statistical Software: Release 15. College Station, TX: StataCorp LLC, 2017). We compiled a data book and classified it with all relevant variables providing simple descriptive statistics with means, proportions, and frequency distributions of NCD risk factors. We calculated the prevalence of the different risk factors for NCDs with 95% confidence intervals, according to sex and age group (18–44, 45–69, and >70). Furthermore, we analysed behavioural and metabolic risk factor data in smaller age-groups to understand patterns among young people, according to weight status (overweight, BMI ≥ 25 or obese, BMI ≥ 30). P-values for comparisons between age groups, weight status or gender were calculated using Chi-square or Fisher's exact test, depending on the distribution and expected frequencies of the data.

### Ethics statement

This study was conducted in full compliance with national regulations and the ethical principles outlined in the Declaration of Helsinki. The protocol was approved by the Universidad Miguel Hernández (UMH) project evaluation board (registration number 2018.291.E.OEP), the nationally accredited ethics board of the Pontificia Universidad Católica de Ecuador (PUCE, reference 2019–27-MB), and the European Research Council Executive Agency (ERCEA, Ref. Ares (2018) 5827042–14/11/2018). All participants provided written informed consent, with the clear option to withdraw from the study or laboratory tests at any time without providing a reason. Signed consent forms were scanned and securely stored in accordance with anonymisation principles to protect participant identity.

## Results

### Population characteristics (Table 1)

Of the 656 participants, 65.4% were women (N = 429) and 34.6% were men (N = 227). The majority were concentrated in the age ranges of 18–44 and 45–69, only a 13.2% were over 70 years old (N = 87). Regarding education, 71,2% (N = 467) of the participants completed at least secondary school (66.6, N = 151 for men and 73.7, N = 316 for women) showing that women had lower education level than men (p = 0.002). Regarding ethnicity, 92% of the participants self-identified as mestizo (N = 604). Over half of participants live with someone (55.6%, N = 365). In terms of employment status, 34.2%

**Table 1. Sociodemographic characteristics of study participants by sex.**

|  | Men | Women | Total |  |
|---|---|---|---|---|
|  | N(%) | N(%) | N(%) | p |
| **Age** |  |  |  |  |
| 18-44 | 93 (41.0) | 193 (45.0) | 286 (44.4) | 0.551 |
| 45-69 | 102 (44.9) | 184 (42.9) | 286 (42.4) |  |
| >70 | 32 (14.1) | 52(12.1) | 84 (13.2) |  |
| **Education** |  |  |  |  |
| No formal Schooling | 2 (0.9) | 25 (5.8) | 27 (4.1) | **0.002** |
| Primary school | 48 (21.2) | 114 (26.6) | 162 (24.7) |  |
| Secondary school | 101 (44.5) | 177 (41.3) | 278 (42.4) |  |
| Technical studies | 19 (8.4) | 41 (9.6) | 60 (9.2) |  |
| University studies | 57 (25.0) | 71 (16.5) | 128 (19.5) |  |
| No response | 0 (0.0) | 1 (0.2) | 1 (0.1) |  |
| **Ethnicity** |  |  |  |  |
| African descent/black | 5 (2.2) | 5 (1.2) | 10 (1.5) | 0.700 |
| Indigenous | 5 (2.2) | 11 (2.6) | 16 (2.4) |  |
| Mestizo | 210 (92.5) | 394 (91.8) | 604 (92.1) |  |
| White | 3 (1.3) | 12 (2.8) | 15 (2.3) |  |
| Other | 3 (1.3) | 6 (1.4) | 9 (1.4) |  |
| No response | 1 (0.5) | 1 (0.2) | 2 (0.3) |  |
| **Marital status** |  |  |  |  |
| Single | 69 (30.4) | 108 (25.2) | 177 (27.0) | 0.366 |
| Married | 104 (45.8) | 190 (44.3) | 294 (44.8) |  |
| Divorced | 21 (9.3) | 49 (11.4) | 70 (10.7) |  |
| Civil union | 22 (9.7) | 49 (11.4) | 71 (10.8) |  |
| Widowed | 11 (4.8) | 33 (7.7) | 44 (6.7) |  |
| **Employment** |  |  |  |  |
| Public sector employee | 18 (7.9) | 22 (5.1) | 40 (6.1) | **<0.001** |
| Private sector employee | 47 (20.7) | 60 (14.0) | 107 (16.3) |  |
| Self-employed* | 90 (39.7) | 134 (31.2) | 224 (34.2) |  |
| Homemakers | 0 (0.0) | 137 (31.9) | 137 (20.9) |  |
| Retired | 32 (14.2) | 16 (3.7) | 48 (7.3) |  |
| Students | 23 (10.1) | 31 (7.2) | 54 (8.2) |  |
| Unemployed able to work | 13 (5.7) | 26 (6.1) | 39 (5.9) |  |
| Unemployed not able to work | 4 (1.7) | 3 (0.8) | 7 (1.1) |  |
| **Monthly income** |  |  |  |  |
| [$0-400) | 47(25.0) | 165(48.0) | 212(39.9) | **<0.001** |
| [$400-800) | 92(48.9) | 140(40.7) | 232(43.6) |  |
| ≥$800 | 49(26.1) | 39(11.3) | 88(16.5) |  |
| **Total** | 227 (100) | 429 (100) | 656 (100) |  |

*self-employment included those who provide services such as a taxi-driver or street vendors

P-values were calculated using the Chi-square ($\chi^2$) test; Fisher's exact test was applied when more than 20% of expected cell counts were less than 5.

were self-employed with significant gender differences (39.7% for men and 31.2% for women, p < 0.001). Overall, 21% (n = 137) of the people interviewed were non-paid homemakers, all of them women, 22.4% were formally employed (6.1% in the public and 16.3% in private sector) and the unemployment rate was 6%. Regarding monthly income, nearly 40% of participants were living on less than 400 dollars per month, a situation that was more frequent among women (p < 0.001).

### Behavioural risk factors (Table 2)

There were significant differences in smoking behaviour, and alcohol consumption between men and women (p < 0.001). Twenty-five per cent (N = 54) of the men surveyed were smokers (daily or occasional) compared to only a 5% of women (N = 22). Table S2 (S1 File) in supplementary material presents a detailed analysis of participants' age which shows that, even though no differences in tobacco consumption by age were observed, 20% of young adults [18–24,29] were smokers, and this behaviour showed a decreasing trend with increasing age. Half of participants (50.8%, N = 333) currently consumed alcohol, and men showed a higher consumption than women (p < 0.001). Overall, young adults [18–24,28] showed the lowest cessation of drinking behaviour (16.05%, N = 13), while early adults [18–25]; [25–35]; [35–60]; >60 had the highest prevalence of alcohol consumption among age groups (p = 0.007, 56.73%, N = 59). A wider age description and alcohol use are provided in S2 (S1 File).

Regarding dietary habits, young and early adults (18-35 years old) showed higher rates of fruit and vegetable consumption, while this behaviour decreased as age increased until older adults (> 60), who presented the lowest adherence of WHO recommendation (p = 0.043). Across all age groups, more than 70% of those surveyed reported high sugar consumption (N = 475). In terms of physical activity, a quarter of those surveyed (24.4%, n = 160) reported engaging in under 150 minutes of physical activity per week. Men appeared to be more physically active than women, with 186 (81.9%) reporting more than 150 minutes of physical activity per week compared to 310 (72.3%) of women (p = 0.006). Differences within men´s age groups were observed (p = 0.012), showing that the prevalence of physical activity decreased abruptly in men over 60, while in women this phenomenon is gradual.

### Metabolic NCD risk factors (Table 3, Fig 2)

More than 70% of individuals were either overweight (41.2%, N = 234) or obese (33.3%, N = 189). Women were more frequently overweight than men (p < 0.001), and among women, those aged 45–69 years were the most affected (p = 0.008, 85.5%, N = 147). Regarding waist circumference, over half of participants (56.8%, N = 322) had an increased waist circumference, with women being more affected than men (60% versus 50.3%, p = 0.001). Hypertension was observed in 26.9% of participants (N = 153) being more common among men compared with women (33.7% versus 23.6% respectively, p = 0.010). In both men and women, hypertension was most frequent among older adults. Table S3 (S1 File) shows an analysis of metabolic risk factors by age-groups and weight status. Individuals inside obesity group were more affected by hypertension (35.4%, N = 59) than those with overweight (25.2%, N = 59, p = 0.022). We observed hypertension in young overweight individuals, (above 5% in those aged 18–35) but this was not observed in obese individuals of the same age. Hyperglycaemia was present in 7.9% of those surveyed (N = 45) with no significant differences by gender. In both men and women, hyperglycaemia increased with by reaching a 12% in men over 70 and 23% in women over 70.

In terms of lipid profile, more than half of the participants had hypercholesterolemia (56.2%, N = 317), with the highest percentage in those aged 70 years and older (70.1%, N = 47). Low protective HDL was found in 55.5% of participants (N = 313), with women presenting higher prevalences than men (p < 0.001, 60.8%, N = 231, 44.6%, N = 82). Remarkably, in those with obesity no differences by age were observed (Table S3 (S1 File)). Nearly half of those surveyed (49.7%, N = 271) had elevated LDL levels. Raised triglycerides were present in 52.8% (N = 298) of participants, with a higher prevalence in men than in women (59.8% versus 49.5 respectively, p = 0.021). In terms of kidney function, elevated serum creatinine was detected in 3% (N = 17) of those surveyed with women more affected than men (p < 0.001, 3.7%, N = 14, 1.6%, N = 3). Half of the participants (52.4%, N = 196) had a cardiovascular disease (CVD) risk of less than 5% in ten years.

**Table 2. Prevalence of behavioural non-communicable disease risk factors by sex and age according to the STEPS survey conducted in the southern district of Quito (2021).**

| Age group | Men (N=227) | | | | | Women (N=429) | | | | | Total (N=656) | | | | | |
|---|---|---|---|---|---|---|---|---|---|---|---|---|---|---|---|---|
| | 18-44 N(%) | 45-69 N(%) | ≥70 N(%) | Overall N(%) | p1 | 18-44 N(%) | 45-69 N(%) | ≥70 N(%) | Overall N(%) | p1 | 18-44 N(%) | 45-69 N(%) | ≥70 N(%) | Overall N(%) | p1 | p2 |
| **Tobacco use** | | | | | | | | | | | | | | | | |
| Daily Smoker | 10(10.8) | 9(8.8) | 2(6.2) | 21(9.3) | 0.187 | 6(3.1) | 2(1.1) | 2(3.9) | 10(2.3) | 0.316 | 16(5.6) | 11(3.8) | 4(4.7) | 31(4.7) | 0.147 | <0.001 |
| Occasional Smoker | 19(20.4) | 12(11.8) | 2(6.3) | 33(14.5) | | 8(4.1) | 3(1.6) | 1(1.9) | 12(2.8) | | 27(9.4) | 15(5.2) | 3(3.6) | 45(6.9) | | |
| Never Smoke | 64(68.8) | 81(79.4) | 28(87.5) | 173(76.2) | | 179(92.7) | 179(97.3) | 49(94.2) | 407(94.9) | | 243(85.0) | 260(90.9) | 77(91.7) | 580(88.4) | | |
| **Alcohol consumption** | | | | | | | | | | | | | | | | |
| Current drinker | 52(55.9) | 52(51.0) | 14(43.7) | 118(52.0) | 0.715 | 97(50.2) | 96(52.2) | 22(42.3) | 215(50.1) | 0.124 | 149(52.1) | 148(51.8) | 36(42.9) | 333(50.8) | 0.421 | <0.001 |
| Ex-drinker | 28(30.1) | 35(34.3) | 14(43.8) | 77(33.9) | | 48(24.9) | 38(20.6) | 8(15.4) | 94(21.9) | | 76(26.6) | 73(25.5) | 22(26.2) | 171(26.1) | | |
| Never Drink | 13(14.0) | 15(14.7) | 4(12.5) | 32(14.1) | | 48(24.9) | 50(27.2) | 22(42.3) | 120(28.0) | | 61(21.3) | 65(22.7) | 26(30.9) | 152(23.1) | | |
| **Fruit and Vegetables consumption** | | | | | | | | | | | | | | | | |
| <5 portions per day | 76(81.7) | 94(92.2) | 29(90.6) | 199(87.7) | 0.082 | 168(87.1) | 168(91.3) | 45(86.5) | 381(88.8) | 0.343 | 244(85.3) | 262(91.6) | 74(88.1) | 580(88.4) | 0.063 | 0.663 |
| ≥5 portions per day | 17(18.3) | 8(7.8) | 3(9.4) | 28(12.3) | | 25(12.9) | 16(8.7) | 7(13.4) | 48(11.2) | | 42(14.7) | 24(8.4) | 10(11.9) | 76(11.6) | | |
| **Salt consumption** | | | | | | | | | | | | | | | | |
| Low | 59(63.4) | 70(68.6) | 19(59.4) | 148(65.2) | 0.555 | 124(64.3) | 129(70.1) | 37(71.2) | 290(67.6) | 0.419 | 183(64.0) | 199(69.6) | 56(66.7) | 438(66.8) | 0.365 | 0.535 |
| High | 34(36.6) | 32(31.4) | 13(40.6) | 79(34.8) | | 69(35.7) | 55(29.9) | 15(28.8) | 139(32.4) | | 103(36.0) | 87(30.4) | 28(33.3) | 218(33.2) | | |
| **Sugar consumption** | | | | | | | | | | | | | | | | |
| Low | 29(31.2) | 25(24.5) | 10(31.2) | 64(28.2) | 0.527 | 46(23.8) | 56(30.4) | 15(28.8) | 117(27.3) | 0.334 | 75(26.2) | 81(28.3) | 25(29.8) | 181(27.6) | 0.763 | 0.802 |
| High | 64(68.8) | 77(75.5) | 22(68.8) | 163(71.8) | | 147(76.2) | 125(69.6) | 37(71.2) | 312(72.7) | | 211(73.8) | 205(71.7) | 59(70.2) | 475(72.4) | | |
| **Physical activity per week** | | | | | | | | | | | | | | | | |
| <150 mins | 15(16.1) | 14(13.7) | 12(37.5) | 41(18.1) | 0.012 | 52(26.9) | 46(25.0) | 21(40.4) | 119(27.7) | 0.095 | 67(23.4) | 60(21.0) | 33(39.3) | 160(24.4) | 0.003 | 0.006 |
| ≥150 mins | 78(83.9) | 88(86.3) | 20(62.5) | 186(81.9) | | 141(73.1) | 138(75.0) | 31(59.6) | 310(72.3) | | 219(76.6) | 226(79.0) | 51(60.7) | 496(75.6) | | |

P[1]-values comparing age-groups, P[2] values comparing overall prevalences by gender. All P values were calculated using the Chi-square (χ²) test; Fisher's exact test was applied when more than 20% of expected cell counts were less than 5.

**Table 3. Prevalence of metabolic non-communicable disease risk factors by sex and age according to the STEPS survey conducted in the southern district of Quito (2021).**

| | Men | | | | | Women | | | | | Total | | | | | |
|---|---|---|---|---|---|---|---|---|---|---|---|---|---|---|---|---|
| Age group | 18-44 | 45-69 | >70 | Overall | p¹ | 18-44 | 45-69 | >70 | Overall | p¹ | 18-44 | 45-69 | >70 | Overall | p¹ | p² |
| | N(%) | N(%) | N(%) | N(%) | | N(%) | N(%) | N(%) | N(%) | | N(%) | N(%) | N(%) | N(%) | | |
| **BMI** | | | | N=186 | 0.249 | | | | N=382 | 0.008 | | | | N=568 | 0.009 | <0.001 |
| (<18.5kg/m2) | 2(2.9) | 0(0.0) | 0(0.0) | 2(1.1) | | 1(0.6) | 0(0.0) | 0(0.0) | 1(0.3) | | 3(1.3) | 0(0.0) | 0(0.0) | 3(0.5) | | |
| (18.5–24.9kg/m2) | 24(34.3) | 29(31.5) | 7(29.2) | 60(32.3) | | 49(29.3) | 25(14.5) | 8(18.6) | 82(21.5) | | 73(30.8) | 54(20.5) | 15(22.4) | 142(25.0) | | |
| (25.0–29.9kg/m2) | 29(41.4) | 39(42.4) | 15(62.5) | 83(44.6) | | 65(38.9) | 67(39.0) | 19(44.2) | 151(39.5) | | 94(39.6) | 106(40.1) | 34(50.7) | 234(41.2) | | |
| (≥30kg/m2) | 15(21.4) | 24(26.1) | 2(8.3) | 41(22.0) | | 52(31.1) | 80(46.5) | 16(37.2) | 148(38.7) | | 67(28.3) | 104(39.4) | 18(26.9) | 189(33.3) | | |
| **Waist circumference** | | | | N=187 | 0.021 | | | | N=380 | <0.001 | | | | N=567 | <0.001 | 0.028 |
| Normal | 44(62.9) | 38(41.3) | 11(44.0) | 93(49.7) | | 93(56.0) | 49(28.7) | 10(23.3) | 152(40.0) | | 137(58.1) | 87(33.1) | 21(30.9) | 245(43.2) | | |
| Raised | 26(37.1) | 54(58.7) | 14(56.0) | 94(50.3) | | 73(44.0) | 122(71.3) | 33(76.7) | 228(60.0) | | 99(41.9) | 176(66.9) | 47(69.1) | 322(56.8) | | |
| **Blood pressure status** | | | | N=187 | <0.001 | | | | N=382 | <0.001 | | | | N=569 | <0.001 | 0.010 |
| Normotension | 61(87.1) | 52(56.5) | 11(44.0) | 124(66.3) | | 158(94.6) | 119(69.2) | 15(34.9) | 292(76.4) | | 219(92.4) | 171(64.7) | 26(38.2) | 416(73.1) | | |
| Hypertension | 9(12.9) | 40(43.4) | 14(56.0) | 63(33.7) | | 9(5.4) | 53(30.8) | 28(65.1) | 90(23.6) | | 18(7.6) | 93(35.3) | 42(61.8) | 153(26.9) | | |
| **Blood glucose status** | | | | N=186 | 0.002 | | | | N=383 | 0.002 | | | | N=569 | <0.001 | 0.857 |
| Normoglycemia | 61(88.4) | 55(61.1) | 16(64.0) | 134(72.0) | | 152(90.5) | 103(59.9) | 19(44.2) | 274(71.5) | | 213(89.9) | 160(60.6) | 35(51.5) | 408(71.7) | | |
| Impaired tolerance | 6(8.7) | 27(30.0) | 6(24.0) | 39(21.0) | | 14(8.3) | 49(28.5) | 14(32.6) | 77(20.1) | | 20(8.4) | 76(28.8) | 20(29.4) | 116(20.4) | | |
| Hyperglycaemia | 2(2.9) | 8(8.9) | 3(12.0) | 13(7.0) | | 2(1.2) | 20(11.6) | 10(23.2) | 32(8.4) | | 4(1.7) | 28(10.6) | 13(19.1) | 45(7.9) | | |
| **Cholesterol status** | | | | N=184 | 0.023 | | | | N=380 | <0.001 | | | | N=564 | <0.001 | 0.916 |
| Normal | 38(55.9) | 35(38.5) | 7(28.0) | 80(43.5) | | 104(61.9) | 50(29.4) | 15(31.0) | 167(44.0) | | 142(60.2) | 85(32.6) | 20(29.9) | 247(43.8) | | |
| Hypercholesterolemia | 30(44.1) | 56(61.5) | 18(72.0) | 104(56.5) | | 64(38.1) | 120(70.6) | 29(69.0) | 213(56.0) | | 94(39.8) | 176(67.4) | 47(70.1) | 317(56.2) | | |
| **HDL** | | | | N=184 | 0.051 | | | | N=380 | 0.035 | | | | N=564 | 0.001 | <0.001 |
| Lower | 38(55.9) | 36(39.6) | 8(32.0) | 82(44.6) | | 113(67.3) | 98(57.6) | 20(47.6) | 231(60.8) | | 151(64.0) | 134(51.3) | 28(41.8) | 313(55.5) | | |
| Normal | 30(44.1) | 55(60.4) | 17(68.0) | 102(55.4) | | 55(32.7) | 72(42.4) | 22(52.4) | 149(39.2) | | 85(36.0) | 127(48.7) | 39(58.2) | 251(44.5) | | |
| **LDL** | | | | N=177 | 0.214 | | | | N=368 | <0.001 | | | | N=545 | <0.001 | 0.853 |
| Normal | 39(59.1) | 41(47.1) | 10(41.7) | 90(50.8) | | 110(67.1) | 56(34.6) | 18(42.9) | 184(50.0) | | 149(64.8) | 97(39.0) | 28(42.4) | 274(50.3) | | |
| Raised | 27(40.9) | 46(52.9) | 14(58.3) | 87(49.2) | | 54(32.9) | 106(65.4) | 24(57.1) | 184(50.0) | | 81(35.2) | 152(61.0) | 38(57.6) | 271(49.7) | | |

*(Continued)*

**Table 3.** (Continued)

| Age group | Men 18-44 | 45-69 | >70 | Overall | P | Women 18-44 | 45-69 | >70 | Overall | P | Total 18-44 | 45-69 | >70 | Overall | P | P² |
|---|---|---|---|---|---|---|---|---|---|---|---|---|---|---|---|---|
| **Triglycerides** | N=184 | | | | | | | | N=380 | | | | | N=564 | | |
| Normal | 27(39.7) | 33(36.3) | 14(56.0) | 74(40.2) | 0.211 | 107(63.7) | 69(40.6) | 16(38.1) | 192(50.5) | <0.001 | 134(56.8) | 102(39.1) | 30(44.8) | 266(47.2) | <0.001 | 0.021 |
| Raised | 41(60.3) | 58(63.7) | 11(44.0) | 110(59.8) | | 61(36.3) | 101(59.4) | 26(61.9) | 188(49.5) | | 102(43.2) | 159(60.9) | 37(55.2) | 298(52.8) | | |
| **Seric Creatinine** | | | | N=184 | | | | | N=380 | | | | | N=564 | | |
| Lower | 3(4.4) | 6(6.6) | 1(4.0) | 10(5.4) | 0.132 | 32(19.0) | 24(14.1) | 4(8.9) | 60(15.8) | 0.011 | 35(14.8) | 30(11.5) | 5(7.5) | 70(12.4) | <0.001 | 0.009 |
| Normal | 64(94.1) | 85(93.4) | 22(88.0) | 171(92.9) | | 135(80.4) | 137(80.6) | 37(82.2) | 306(80.5) | | 199(84.3) | 222(85.1) | 56(83.5) | 477(84.6) | | |
| Raised | 1(1.5) | 0(0.0) | 2(8.0) | 3(1.6) | | 1(0.6) | 9(5.3) | 4(8.9) | 14(3.7) | | 2(0.9) | 9(3.4) | 6(9.0) | 17(3.0) | | |
| **Age group >40** | 40-54 | 55-69 | >70 | Overall | | 40-54 | 55-69 | >70 | Overall | | 40-54 | 55-69 | >70 | Overall | | |
| | | | | N=125 | | | | | N=249 | | | | | N=374 | | |
| **Cardiovascular Risk** | | | | | | | | | | | | | | | | |
| <5% | 37(80.4) | 10(18.5) | 0(0.0) | 47(37.6) | <0.001 | 94(97.9) | 55(49.5) | 0(0.0) | 149(59.8) | <0.001 | 131(92.3) | 65(39.4) | 0(0.0) | 196(52.4) | <0.001 | <0.001 |
| 5–10% | 9(19.6) | 32(59.3) | 4(16.0) | 45(36.0) | | 2(2.1) | 51(46.0) | 22(52.4) | 75(30.1) | | 11(7.7) | 83(50.3) | 26(38.8) | 120(32.1) | | |
| 10–20% | 0(0.0) | 11(20.4) | 20(80.0) | 31(24.8) | | 0(0.0) | 5(4.5) | 19(45.2) | 24(9.6) | | 0(0.0) | 16(9.7) | 39(58.2) | 55(14.7) | | |
| 20–30% | 0(0.0) | 1(1.8) | 1(4.0) | 2(1.6) | | 0(0.0) | 0(0.0) | 1(2.4) | 1(0.4) | | 0(0.0) | 1(0.6) | 2(3.0) | 3(0.8) | | |
| >30% | 0(0.0) | 0(0.0) | 0(0.0) | 0(0.0) | | 0(0.0) | 0(0.0) | 0(0.0) | 0(0.0) | | 0(0.0) | 0(0.0) | 0(0.0) | 0(0.0) | | |

P¹values comparing age-groups, P² values comparing overall prevalences by gender. All P values were calculated using the Chi-square ($\chi^2$) test; Fisher's exact test was applied when more than 20% of expected cell counts were less than 5.

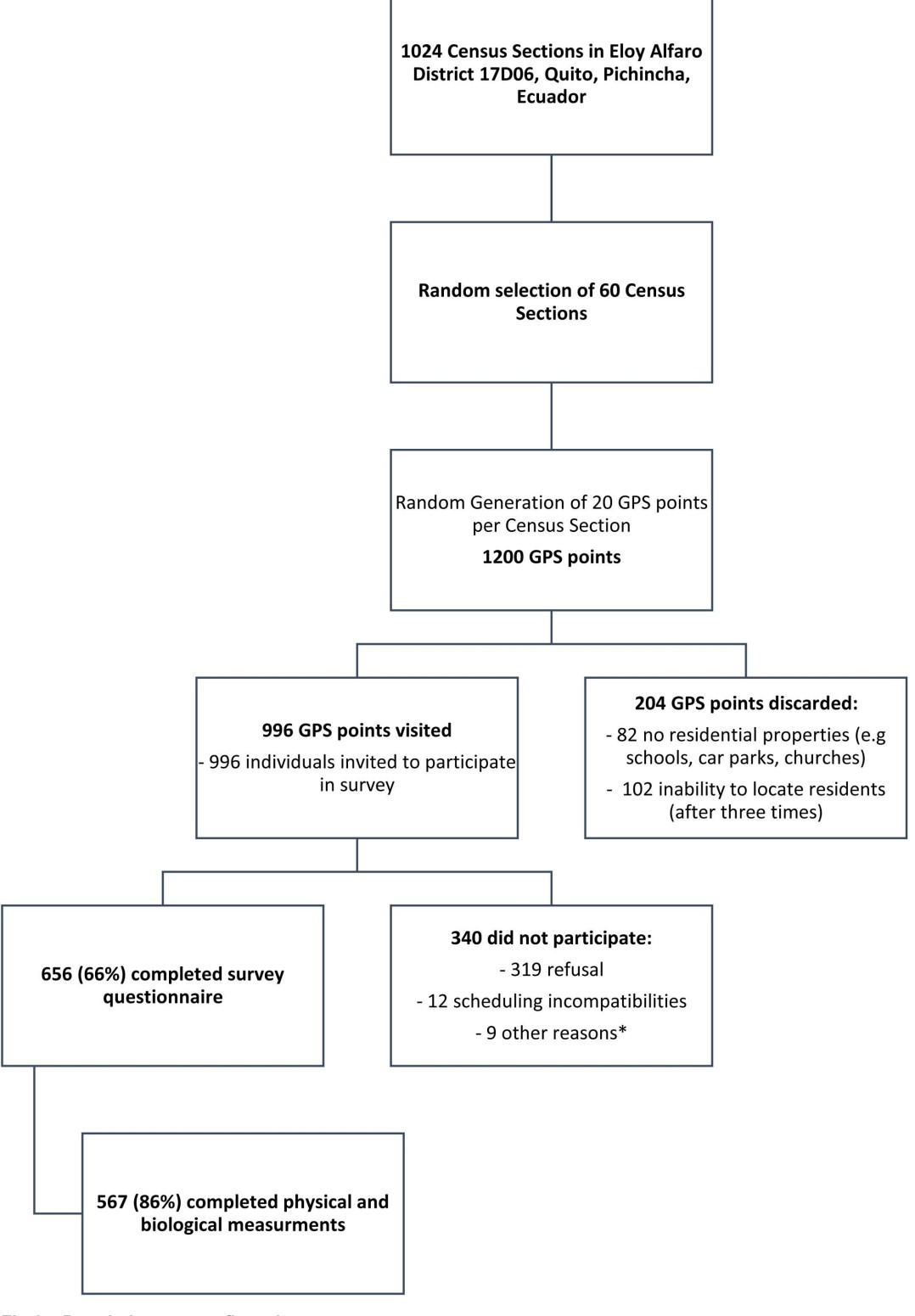

**Fig 2. Population survey flow chart.**

The prevalence of CVD risk over 10% was higher in men (26.48%, N = 331) than in women (10.96%, N = 254; p < 0.001). We noted the presence of individuals with significantly higher values, such as those who reported a BMI over 40 (3.17%, N = 18), fasting blood glucose levels above 300 mg/dL (1.41%, N = 8), or blood pressure readings over 160 (1.41%, N = 8) systolic and 110 diastolic (0.88%, N = 5) (Figs 3).

## Discussion

The obesity pandemic significantly affects the South American region, making it one of the most impacted locations in the world [30,31]. Our study highlights overweight and obesity as an alarming public health issue in a low-income urban district of Quito, affecting more than 70% of this urban population. These conditions, associated with many of the other NCD risk factors, are manifestations of multicausal complex metabolic disorders and are directly related to an unhealthy diet, low physical activity, and high sugar consumption, which may link with how the food industry [32] and commercial determinants of health influence individual behaviours and consumption patterns [33]. Increasing consumption of ultra-processed foods (high in sugar, salt, and fat) and alcoholic drinks has been observed in Ecuador and other Latin American countries across all ages and social classes, but particularly affecting populations with lower socio-economic status [34].

In 2016, the Latin American Federation of Obesity Societies (FLASO) highlighted Ecuador as the Latin American region with the lowest obesity prevalence (14.2%), yet they stressed that recent reports pointed to increasing rates [35]. The values reported here are marginally higher than those found in the STEPwise survey carried out in 2018 [16] which reported rates of 37.9% for overweight and rates of 25.7% for obesity. Similarly, in the same year, the national survey of Health and Nutrition (ENSANUT) in Ecuador [36], reported that overweight and obesity affected nearly 64.7% of the population. In addition, our data from a low-income neighbourhood show a higher prevalence of obesity in women than in men and differences in metabolic NCDs risk factors. Traditional gender patterns in health-related behaviours such as tobacco and alcohol consumption and physical activity were also observed. Gender differences should be addressed by policymakers and public health strategies, recognising the link between gender roles and gender-based imposition of home and care responsibilities prevalent in Latin American culture [37].

Metabolic risk factors, such as hypertension, raised glucose status, and hypercholesterolemia, are widely influenced by overweight and obesity. The prevalence of hypertension found in our study was similar to results found in other studies carried out in the Latin America region [38–40]. Although other countries with similar characteristics, such Venezuela [41], Colombia [42], or Peru [43] had higher rates than ours, it is important to note that the average prevalence of hypertension at altitude in Latin America and the Caribbean was 19.1% [38]. The prevalence of hypertension in the south of Quito is one-third higher than the average for regions with similar characteristics, which may indicate a role for lifestyles factors influenced by urbanisation and high pollution [38,44,45]. However, it is significantly lower than found in other Latin American and Caribbean populations (39.1%) [46].

Regarding hyperglycaemia, our findings (7.9%) are significantly higher than other reported prevalences in Ecuador. The CARMELA study indicated 26 years ago that the prevalence of T2DM in Quito was approximately 6.2% of those surveyed [47]. Similar research conducted in in the norther region of Ecuador, Esmeraldas, found a 6.8% prevalence of diabetes with marked differences according to sex [37]. A recent review stated that the prevalence of T2DM in Ecuador was approximately 5.2% [48], with similar rates presented by the Diabetes Atlas (2021) for Ecuador (5.5%) [49] . Although our results do not show significant differences in glucose status by gender, literature widely describes diabetes being more prevalent among women than men [50], and higher diabetes related-mortality rates for women [51].

Our results show that more than half of those surveyed were affected by hypercholesterolemia and hypertriglyceridemia, which are significantly higher than figures reported in similar research [52–54]. A 2020 review indicated that across 12 studies conducted in Latin America, the prevalence of raised triglycerides was approximately 43% [54]. Unhealthy behavioural patterns related to massive urbanisation, such as poor diet, lack of green spaces, and air or water pollution could be playing a central role in this higher prevalence rates [55].

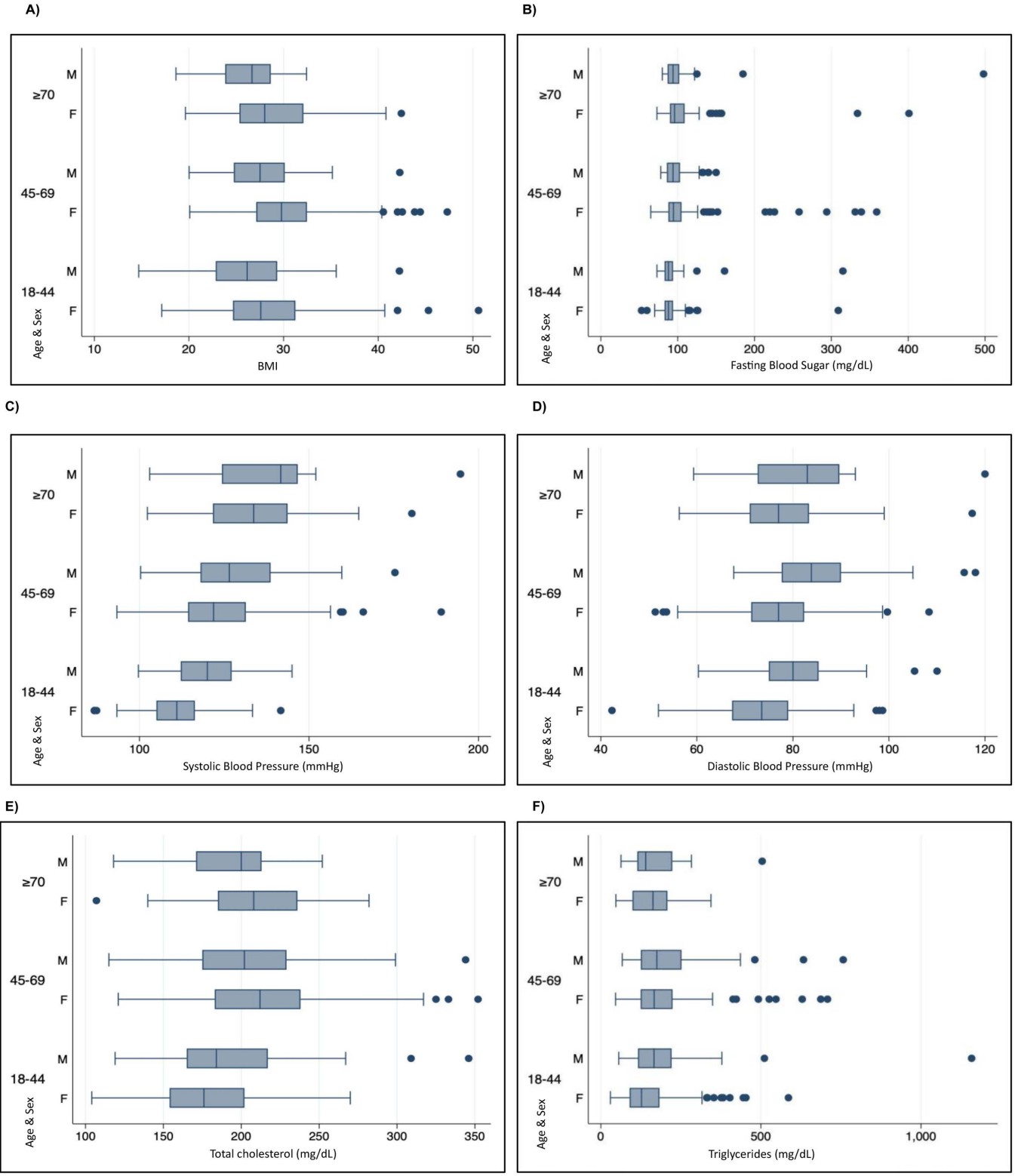

**Fig 3. Metabolic non-communicable disease risk factors compared by participants´ age (years) and sex (M = Male, F = Female).**

It is important to stress that we used the WHO's standardised STEPSwise survey tool and we observed some ambiguity in questions relating to behaviours such as smoking or alcohol consumption. For instance, a person who stopped drinking alcohol six months prior is still classified as a drinker. This factor may contribute to information bias. As noted in the methods section, this survey was conducted during the COVID-19 pandemic, and refusal rates were higher than expected due to participants' fear of engaging in the study. This reflects the challenges of conducting health research during a pandemic [56]. We acknowledge that it would have been recommendable to collect more detail about the reasons for non-participation. We only retained some field notes in which interviewers qualitatively recorded that, in some cases, participants simply did not open the door due to concerns about insecurity. We emphasise this limitation because the prevalence results may be affected. Our study aimed to describe the health status of a low-income context, often characterised by hidden and inaccessible populations, who may be among those who refused participation. Consequently, we may be underestimating alarming prevalence rates.

## Conclusions

This descriptive study reveals an alarmingly high prevalence of overweight and obesity in the study district, accompanied by low consumption of fruit and vegetables and high consumption of sugary products, and directly linked with NCD metabolic risk factors. Effective public health interventions should adopt ecological approaches that address the underlying socioeconomic and environmental determinants. As such, in contexts such as southern Quito, where poverty, limited access to health services, and the availability of unhealthy food options constrain behavioural change, strategies must go beyond individual-focused recommendations.

Synergistic public health efforts that improve both access to healthy foods and knowledge about nutrition, supported by health professionals and strong community networks, can effectively strengthen health promotion and NCD prevention.

## Supporting information

**S1 File.   Table S1**. Definitions and cut-off points of the analysed behavioural, metabolic, and ardiovascular risk variables. **Table S2**. Prevalence of behavioural non-communicable disease risk factors by age. **Table S3**. Prevalence of metabolic non-communicable disease risk factors by age and weight status.
(DOCX)

**S2 File.  Encuesta-STEPS-Preguntas.**
(PDF)

## Acknowledgments

Authors would like to thank the CEAD researcher team who were implicated in different parts of the process from facilitating the authorisation process with local authorities (Montalvo G), developing the field work (Hernández M., Banazizi-Dahbi I.), and data depuration Ocaña Navas, J.A., Grijalva Narvaez, D.F.).

## Author contributions

**Conceptualization:** Sergio Morales-Garzón, Juan Vasconez, Francisco Barrera-Guarderas, Elisa Chilet-Rosell, Lucy Anne Parker.

**Data curation:** Sergio Morales-Garzón, Jessica Pinto Delgado, Lucy Anne Parker.

**Formal analysis:** Sergio Morales-Garzón, Juan Vasconez, Francisco Barrera-Guarderas, Lucy Anne Parker.

**Funding acquisition:** Lucy Anne Parker.

**Investigation:** Sergio Morales-Garzón, Jessica Pinto Delgado, Lucy Anne Parker.

**Methodology:** Sergio Morales-Garzón.

**Project administration:** Jessica Pinto Delgado.

**Resources:** Jessica Pinto Delgado.

**Supervision:** Juan Vasconez, Francisco Barrera-Guarderas, Elisa Chilet-Rosell, Lucy Anne Parker.

**Writing – original draft:** Sergio Morales-Garzón, Juan Vasconez, Lucy Anne Parker.

**Writing – review & editing:** Sergio Morales-Garzón, Juan Vasconez, Francisco Barrera-Guarderas, Elisa Chilet-Rosell, Marta Puig-García, Andrés Peralta, María Fernanda Rivadeneira Guerrero, Ana Lucía Torres-Castillo, Lucy Anne Parker.

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
