## [Decision Letter · Decision Letter 0]

15 May 2025

PONE-D-24-46891The Burden of Non-Communicable Disease Risk Factors in a Low-Income Population: Findings from a Cross-Sectional Study Highlighting the Prevalence of Obesity, Hypertension, and Metabolic Disorders in the south of Quito, Ecuador.PLOS ONE

Dear Dr. Morales-Garzón,

Thank you for submitting your manuscript to PLOS ONE. After careful consideration, we feel that it has merit but does not fully meet PLOS ONE’s publication criteria as it currently stands. Therefore, we invite you to submit a revised version of the manuscript that addresses the points raised during the review process.

We look forward to receiving your revised manuscript.

Kind regards,

José M. Alvarez-Suarez

Academic Editor

PLOS ONE

Journal Requirements:

“This project has received funding from the European Research Council (ERC) under the European Union’s Horizon 2020 research and innovation programme (Grant agreement No. 804761)”

Reviewers' comments:

Reviewer's Responses to Questions

**Comments to the Author**

1. Is the manuscript technically sound, and do the data support the conclusions?

Reviewer #1: Yes

Reviewer #2: Partly

2. Has the statistical analysis been performed appropriately and rigorously? 

Reviewer #1: No

Reviewer #2: No

3. Have the authors made all data underlying the findings in their manuscript fully available?

Reviewer #1: Yes

Reviewer #2: Yes

4. Is the manuscript presented in an intelligible fashion and written in standard English?

Reviewer #1: Yes

Reviewer #2: Yes

5. Review Comments to the Author

Reviewer #1: The authors are grateful for presenting a research paper in a Latin American country. The introduction considers the international contextual aspects as appropriate, emphasizing the lack of data in LAC. The aim was to describe the prevalence of risk factors for chronic diseases in a low-income district in southern Quito. It is not clear why this area is being studied.

A better characterization of the study site in terms of environmental, meteorological, and social aspects would be helpful; a map would be helpful. It is recommended that Figure 1 detail the characteristics of the GPS points that were eliminated and the participants who refused to participate.

The Recruitment and Procedures section specifies that the cultural adaptations made to the questionnaires be detailed. In line 157, the criteria for eligible participants for the glucose tolerance test are not clear. References for the section are missing between lines 160 and 164.

It is essential to complete the data analysis section. It is not clear why multivariate analyses are not performed to identify risk factors associated with the higher prevalence rates determined.

In Tables 1, 2, and 3, it is suggested to add whether the differences according to sex are significant or not, detailing the hypothesis tests used.

In the discussion section, other aspects that could be related to the determined prevalence rates are not addressed. What is the importance of the environmental characteristics that may be related to the chronic diseases indicated? For example, what is the role of altitude and air pollution?

The conclusions do not state possible interventions or public health actions that would allow the prevalence rates obtained to be reduced.

This type of research must be published, so it is hoped that the authors will be able to present an improved version as soon as possible.

Reviewer #2: The authors present a study on the prevalence of CNCD risk factors in a low-income population. While the STEP methodology, applied nationwide in Ecuador (2018), already incorporated stratification by area of residence (urban/rural), which presumably encompassed low-resource populations, the manuscript should more clearly justify the rationale for this specific study. What particular aspects of the investigated low-income population were not adequately addressed by the national STEPS study? Questions and suggestions for the manuscript are presented below.

- Although the findings from the sample (n=656) are consistent with the data from the national sample (n=4,638), their capacity to contribute novel information appears limited. It is suggested that the analysis of sociodemographic characteristics be reoriented towards greater contextual relevance. Instead of the sex-based distribution presented in Table 1, considering the distribution by age and educational attainment could reveal significant inequalities. Furthermore, analyzing the interplay between educational attainment, employment status (employed/unemployed), and gender would facilitate a more comprehensive understanding of labor inclusion challenges within this setting. Isolated data, such as the percentage of female employment in the public sector, without stratification by educational level, lacks the necessary depth to inform context-sensitive intervention strategies for low-resource settings.

This is even more so when their conclusions state the following: "The established links between hypercholesterolaemia, diabetes, and hypertension with unhealthy patterns and pathogenic environments suggest that actions should be defined to address the social determinants of health, incorporating insights from various fields to reduce inequalities, particularly gender inequalities, and create fair environments that promote and improve health of men and women".

- It would be valuable to consider age stratification with ranges allowing for enhanced granularity in the prevalence analysis, particularly within groups exhibiting dynamic consumption patterns (Table 2). For instance, disaggregating age cohorts into Young Adults (18-25 y), Early Adults (25-35 y), Middle-Aged Adults (35-60 y), and Older Adults (60 years and older) would be beneficial. Specifically, further disaggregation of the 18-40 year range into narrower subgroups, such as 18-25 years, could elucidate specific trends in young adulthood, a stage where, as noted, consumption is often influenced by autonomy, university/work life, and social determinants, including risky behaviors like binge drinking and social smoking.

- To enhance the interpretation of metabolic risk factor outcomes (Table 3), additional stratification by nutritional status, beyond age cohorts, would be valuable. As evidenced by prior research, including recent studies in the Ecuadorian population lipid profiles and other metabolic parameters can vary significantly according to nutritional status. Incorporating this stratification could facilitate the identification of specific subpopulations within age ranges exhibiting distinct metabolic risk profiles, thereby informing more targeted interventions.

Ref: Albuja Quintana, Natalia (2025) Relationship between plasma uric acid levels, antioxidant capacity, and oxidative damage markers in overweight and obese adults: A cross-sectional study. https://doi.org/10.1371/journal.pone.0312217

In the discussion section, the authors excessively elaborate on mechanisms that their own data cannot substantiate, as exemplified in lines 295-305, 312-318, 328-336, and 341-345. Given the cross-sectional design and the limited scope of statistical analyses, mechanistic explanations are challenging to support without additional experimental evidence. It would be more appropriate for the authors to reorganize the discussion section based on their own findings to clearly articulate the novelties or actual conclusions of this study to the readers.

- Starting on line 73, the bibliographic citation has a different format that should be corrected, from ref 12 to ref 17.

- Line 341: The information is repeated

These findings highlight. These findings highlight the high burden o…

- Figure 2: Does it not describe the units of measurement for each parameter, for example, total cholesterol is in mg/dl?

6. PLOS authors have the option to publish the peer review history of their article (what does this mean? ). If published, this will include your full peer review and any attached files.

**Do you want your identity to be public for this peer review?** For information about this choice, including consent withdrawal, please see our Privacy Policy .

Reviewer #1: **Yes: ** Sandra Cortés

Reviewer #2: No

---

## [Author Response · Author response to Decision Letter 1]

1 Jul 2025

We would like to sincerely thank both reviewers for their time and careful reading of our manuscript and for providing insightful and constructive comments. Your observations and suggestions are valuable and have helped us to improve the quality and clarity of our study.

Reviewer 1: “The authors are grateful for presenting a research paper in a Latin American country. The introduction considers the international contextual aspects as appropriate, emphasizing the lack of data in LAC. The aim was to describe the prevalence of risk factors for chronic diseases in a low-income district in southern Quito. It is not clear why this area is being studied.”

“A better characterization of the study site in terms of environmental, meteorological, and social aspects would be helpful; a map would be helpful.”

This study is part of the European CEAD project, which aims to provide rigorous epidemiologic data on diabetes risk and morbidity in both urban and rural low-income districts of Ecuador. We included a broader description of the CEAD project's objectives in lines 92–94 and the rationale for selecting Quito as the study setting in lines 110–118. We also included a figure (Figure 1) which includes a map of described locations.

It is recommended that Figure 1 detail the characteristics of the GPS points that were eliminated and the participants who refused to participate.

We broadened our description of eliminated points in the figure. We also included in the Methods (lines 142-147) and in discussion (Lines 398-400), that the recruitment was done during the pandemic, which affected how people participated.

The Recruitment and Procedures section specifies that the cultural adaptations made to the questionnaires be detailed.

We included a paragraph in lines 179-183 describing the cultural adaptations. We did not make major changes to the original questionnaire. Instead, we adapted the concepts and the nature of the questions to fit the Ecuador context, as per the STEP guidelines (World Health Organization, 2006). For example, in the sociodemographic section, the educational level scale differs slightly, we also adapted job types considering the Ecuadorian context and we included some visuals items (photos) related to the most common types of fruits and vegetables.

- World Health Organization. Noncommunicable Diseases and Mental Health Cluster. Manual de vigilancia STEPS de la OMS: el método STEPwise de la OMS para la vigilancia de los factores de riesgo de las enfermedades crónicas [Internet]. World Health Organization; 2006 [cited 2023 Feb 7]. Report No.: WHO/NMH/CHP/SIP/05.02. Available from: https://apps.who.int/iris/handle/10665/43580

In line 157, the criteria for eligible participants for the glucose tolerance test are not clear.

We have improved the explanation in lines 189–202 regarding the process for selecting participants prior to the laboratory tests.

References for the section are missing between lines 160 and 164

We have included the reference “Zhou J, Fabros A, Lam SJ, Coro A, Selvaratnam R, Brinc D, Di Meo A. The stability of 65 biochemistry analytes in plasma, serum, and whole blood. Clin Chem Lab Med. 2024 Mar 7;62(8):1557-1569. doi: 10.1515/cclm-2023-1192. PMID: 38443327.” in lines 197 that details the management and treatment of the biological samples.

It is essential to complete the data analysis section. It is not clear why multivariate analyses are not performed to identify risk factors associated with the higher prevalence rates determined. In Tables 1, 2, and 3, it is suggested to add whether the differences according to sex are significant or not, detailing the hypothesis tests used.

We have revised the data analysis section to reflect improvements. We fully agree that the initial analysis was relatively simple aiming to describe the general health condition of participants.

We did not conduct multivariable analyses for each of the risk factors due to the large number of variables (both outcome variables and potential explanatory variables) and the volume of data involved. For examples, we could develop a separate multivariable model for each risk factor (behavioural and metabolic), using the population’s sociodemographic characteristics as potential explanatory variables, but we do not believe it is feasible to incorporate all these analyses within a single paper.

We do agree, however, that this type of analysis is important, and in fact, different members of our international team are currently undertaking such analyses, each focusing on specific research questions. For this study, our primary aim was to describe the prevalence of numerous NCD risk factors in this low-income urban setting, and as such we prioritized a descriptive approach over multivariate analyses. We acknowledge this as a limitation and have clarified it in the discussion.

Nevertheless, we agree that it is important to include p-values in Tables 1, 2, and 3 to indicate the statistical significance of differences observed between variables. We now present the p values for age-group (stratified by sex) and the comparison according to sex, as suggested by the reviewer. The hypothesis tests used are now detailed in the methods section.

In the discussion section, other aspects that could be related to the determined prevalence rates are not addressed. What is the importance of the environmental characteristics that may be related to the chronic diseases indicated? For example, what is the role of altitude and air pollution?

We have included in the discussion section (lines 364–365, 381-383) references that highlight how environmental characteristics—such as low levels of urbanisation, low pollution, and access to green spaces, which are typical of rural high-altitude communities—may protect against chronic diseases. In contrast, in the south of Quito, increasing urbanisation, higher air pollution, and other environmental factors may be contributing to the higher prevalence rates observed.

The conclusions do not state possible interventions or public health actions that would allow the prevalence rates obtained to be reduced.

We have included possible public health interventions in the conclusion (lines 407–423), highlighting the need to address structural barriers like poverty, limited access to specialist health professionals, and unhealthy food availability, while also considering gendered care roles and social inequalities affecting self-care.

Reviewer #2: The authors present a study on the prevalence of CNCD risk factors in a low-income population. While the STEP methodology, applied nationwide in Ecuador (2018), already incorporated stratification by area of residence (urban/rural), which presumably encompassed low-resource populations, the manuscript should more clearly justify the rationale for this specific study. What aspects of the investigated low-income population were not adequately addressed by the national STEPS study?

Thank you for your observation. As we noted in our response to Reviewer 1, our manuscript is part of a broader European-funded project that seeks to describe the burden of risk factors for NCDs with a particular focus on type 2 diabetes mellitus in specific low-income contexts. While we acknowledge that the national STEPS survey conducted in Ecuador in 2018 incorporated urban/rural stratification, it does not provide a picture of urban poor populations. Our study aims to complement national-level data by offering a contextualized analysis that focuses on a specific low-income urban population and offers a detailed presentation of their socioeconomic characteristics. This localized perspective is essential for understanding how NCD risk factors manifest in economically vulnerable contexts. Evaluating local realities can be useful for translating public health recommendations into context-specific actions where standard national data may overlook key local aspects.

At a broader level, our findings may be relevant for researchers, health professionals, and policymakers working in other contexts, as they reveal the frequent presence and coexistence of multiple risk factors within a population characterised by low socioeconomic conditions. This challenges the effectiveness of vertical programs that address individual risk factors in isolation and highlights the potential value of adopting a more ecological and integrated approach, which could lead to greater public health impact. We have broadened our description of the rationale of the study within the context of the CEAD project and provided further details on the study district.

Questions and suggestions for the manuscript are presented below.

- Although the findings from the sample (n=656) are consistent with the data from the national sample (n=4,638), their capacity to contribute novel information appears limited. It is suggested that the analysis of sociodemographic characteristics be reoriented towards greater contextual relevance.

Although our findings are consistent with those of the national survey, we emphasise several alarming patterns that add contextual relevance by highlighting the frequency of NCD risk factors in an urban underserved population.

After conducting univariate and multivariate regression models for each risk factor (behavioural and metabolic), we chose to present a descriptive analysis based on sex and age. This decision was based on our observation that, except in the case of obesity, variables such as education level introduced misleading effects on the results (older adults have more NCD risk factors but they are also the individuals with low level of education, please see the table included at the end of this response). To properly consider the socioeconomic characteristics of the population and how they influence the NCD risk factors, we would need to include a multivariable analysis to consider potential confounding. As mentioned above, in relation to comments from reviewer 1, we did not think it was feasible to include this all in a single manuscript. And in line with the descriptive nature of this study’s objective, and to minimise the risk of misinterpretation, we selected age as the primary variable for the analysis here.

Instead of the sex-based distribution presented in Table 1, considering the distribution by age and educational attainment could reveal significant inequalities. Furthermore, analyzing the interplay between educational attainment, employment status (employed/unemployed), and gender would facilitate a more comprehensive understanding of labor inclusion challenges within this setting. Isolated data, such as the percentage of female employment in the public sector, without stratification by educational level, lacks the necessary depth to inform context-sensitive intervention strategies for low-resource settings.

We believe it is important to keep the sex distribution in Table 1. As the SAGER Guidelines state (doi: 10.1186/s41073-016-0007-6), “data should be routinely presented disaggregated by sex and gender. Sex- and gender-based analyses should be reported regardless of positive or negative outcome.” While we agree that future studies could benefit from a more detailed analysis of the interaction between educational attainment, and employment status, we maintain that presenting the data in Table 1 disaggregated by sex/gender remains relevant.

Evidence suggests that the sex/gender system always intersects with other social axes of inequality in shaping health outcomes. For this reason, our decision to present sex-disaggregated data both in table 1 description of the socioeconomic characteristics of the study population and the NCD risk factors, reflects the study’s scope and the importance of highlighting gendered patterns in NCD risk factors within this underserved urban context.

This is even more so when their conclusions state the following: "The established links between hypercholesterolaemia, diabetes, and hypertension with unhealthy patterns and pathogenic environments suggest that actions should be defined to address the social determinants of health, incorporating insights from various fields to reduce inequalities, particularly gender inequalities, and create fair environments that promote and improve health of men and women".

We have restructured the discussion section to better align our conclusions with the evidence presented in our results. We now highlight specific findings that justify the need for broad population actions that create fairer and more health-promoting environments.

It would be valuable to consider age stratification with ranges allowing for enhanced granularity in the prevalence analysis, particularly within groups exhibiting dynamic consumption patterns (Table 2). For instance, disaggregating age cohorts into Young Adults (18-25 y), Early Adults (25-35 y), Middle-Aged Adults (35-60 y), and Older Adults (60 years and older) would be beneficial. Specifically, further disaggregation of the 18–40-year range into narrower subgroups, such as 18-25 years, could elucidate specific trends in young adulthood, a stage where, as noted, consumption is often influenced by autonomy, university/work life, and social determinants, including risky behaviors like binge drinking and social smoking.

We found your comment highly relevant and fully agree on the importance of enhancing granularity in the analysis, particularly for behavioural risk factors where age in early life can reveal critical patterns. In response, we have included a more detailed analysis in the supplementary material and described prevalences according to the recommended age groups and included references in results. Due to the observed similarities in prevalence between some adjacent groups (e.g. young adults and early adults), we have opted to retain the original age groupings in the main presentation of the manuscript. These groups coincide with those defined in the other STEPS surveys, including the previous Ecuadorian one, which improves comparability of findings.

- To enhance the interpretation of metabolic risk factor outcomes (Table 3), additional stratification by nutritional status, beyond age cohorts, would be valuable. As evidenced by prior research, including recent studies in the Ecuadorian population lipid profiles and other metabolic parameters can vary significantly according to nutritional status. Incorporating this stratification could facilitate the identification of specific subpopulations within age ranges exhibiting distinct metabolic risk profiles, thereby informing more targeted interventions.

Ref: Albuja Quintana, Natalia (2025) Relationship between plasma uric acid levels, antioxidant capacity, and oxidative damage markers in overweight and obese adults: A cross-sectional study. https://doi.org/10.1371/journal.pone.0312217

We found your comment highly valuable. Given that nutritional status emerged as a key result in our analysis, we decided to include an additional table in the supplementary material where overweight and obesity are examined across different age groups. This allows for a more detailed interpretation of metabolic risk factors by age and nutritional status, and supports the identification of subpopulations with distinct risk profiles.

In the discussion section, the authors excessively elaborate on mechanisms that their own data cannot substantiate, as exemplified in lines 295-305, 312-318, 328-336, and 341-345. Given the cross-sectional design and the limited scope of statistical analyses, mechanistic explanations are challenging to support without additional experimental evidence. It would be more appropriate for the authors to reorganize the discussion section based on their own findings to clearly articulate the novelties or actual conclusions of this study to the readers.

We have restructured the discussion section to better reflect the cross-sectional nature of our findings. We have revised the text to ensure that the discussion remains grounded in our own results, focusing on the descriptive nature of the data and clearly outlining the main contributions and novel aspects of the study.

In this regard, the ubiquitous nature of these risk factors also points to the influence of broader

---

## [Decision Letter · Decision Letter 1]

17 Jul 2025

PONE-D-24-46891R1The Burden of Non-Communicable Disease Risk Factors in a Low-Income Population: Findings from a Cross-Sectional Study Highlighting the Prevalence of Obesity, Hypertension, and Metabolic Disorders in the south of Quito, Ecuador.PLOS ONE

Dear Dr. Morales-Garzón,

Thank you for submitting your manuscript to PLOS ONE. After careful consideration, we feel that it has merit but does not fully meet PLOS ONE’s publication criteria as it currently stands. Therefore, we invite you to submit a revised version of the manuscript that addresses the points raised during the review process.

We look forward to receiving your revised manuscript.

Kind regards,

José M. Alvarez-Suarez

Academic Editor

PLOS ONE

Journal Requirements:

Reviewers' comments:

Reviewer's Responses to Questions

**Comments to the Author**

1. If the authors have adequately addressed your comments raised in a previous round of review and you feel that this manuscript is now acceptable for publication, you may indicate that here to bypass the “Comments to the Author” section, enter your conflict of interest statement in the “Confidential to Editor” section, and submit your "Accept" recommendation.

Reviewer #1: (No Response)

Reviewer #2: All comments have been addressed

2. Is the manuscript technically sound, and do the data support the conclusions?

Reviewer #1: Partly

Reviewer #2: Yes

3. Has the statistical analysis been performed appropriately and rigorously? 

Reviewer #1: Yes

Reviewer #2: Yes

4. Have the authors made all data underlying the findings in their manuscript fully available?

Reviewer #1: Yes

Reviewer #2: No

5. Is the manuscript presented in an intelligible fashion and written in standard English?

Reviewer #1: Yes

Reviewer #2: Yes

6. Review Comments to the Author

Reviewer #1: Line 33: Please include details on the sample size and power.

Line 47. Please describe comparisons among the main results and other similar studies in Latin American countries.

Line 51: Add keywords describing the site: Ex, Ecuador, Latin American countries.

Line 110: add indicators about environmental exposures and vulnerability.

Line 127. Figure 1: include latitude and longitude; make edits for improving visibility.

Line 131: Describe the selection criteria for the participants.

Line 147: Review the figure number. Add a food note with details of the 319 refusal participants. Add a table about their descriptive statistic and the implications of the obtained prevalences (biases).

Line 166: Please consider including the questionnaire in the supplementary material.

Line 264: Please clarify if the monetary unitary is dollars or another currency.

Line 345: Add the units for each axis.

General comments

It could be useful to include a list of cut-off points for each biochemical definition.

Use a point to indicate decimal values.

Reviewer #2: (No Response)

7. PLOS authors have the option to publish the peer review history of their article (what does this mean? ). If published, this will include your full peer review and any attached files.

**Do you want your identity to be public for this peer review?** For information about this choice, including consent withdrawal, please see our Privacy Policy .

Reviewer #1: **Yes: ** Sandra Cortés

Reviewer #2: No

---

## [Author Response · Author response to Decision Letter 2]

23 Jul 2025

Dear reviewers, we thank your time and patience on reviewing a second time our manuscript, we are sure that all your reccomendations and advice increase the quality of this research.

Reviewer #1:

Line 33: Please include details on the sample size and power.

We completely agree on the importance of this issue, so we decided to include a detailed description of our sample characteristics in lines 33–35, which states: “We used multi-stage cluster sampling to select 656 of población total adult residents of 17D06 health district, enabling a prevalence estimation with at least ±5.73% absolute precision.” and in lines 136-144 a broader description of the sample method use.

Line 47. Please describe comparisons among the main results and other similar studies in Latin American countries.

We included a phrase in the conclusion section where we compare our results, specifically in lines 48–49, which state: “The critical prevalence of NCD risk factors in this low-income urban district of Quito, alongside similar trends observed in other settings across Latin America, underscores...”

Line 51: Add keywords describing the site: Ex, Ecuador, Latin American countries.

Included

Line 110: add indicators about environmental exposures and vulnerability.

We decided to include some indicator of exposure, impact and poverty segregation characteristichs of Quito in lines 117-127.

Line 127. Figure 1: include latitude and longitude; make edits for improving visibility.

We decided to reestructure the referenced map includign suggestions.

Line 131: Describe the selection criteria for the participants.

Due to the nature of this population survey, all people over 18 resident in the 17D06 district were elegible for the inclusion. Inclusion criteria was age and residence. We included a phrase were we stated it in lines 135

Line 147: Review the figure number. Add a food note with details of the 319 refusal participants. Add a table about their descriptive statistic and the implications of the obtained prevalences (biases).

We acknowledge the importance and the need to describe why some individuals decided not to participate. Unfortunately, when we collected the information in our field notes, the survey team recorded cases of rejection but did not capture detailed reasons. Qualitatively, they reported a high proportion of refusals linked to fear of COVID-19, or individuals who simply did not open the door due to concerns about insecurity. For this reason, we are unable to provide a table as suggested. However, we recognise the limitation this represents, particularly regarding potential biases in the prevalence estimates. We have expanded our description of this limitation in the discussion section in lines 404–408.

Line 166: Please consider including the questionnaire in the supplementary material.

We decided to include the questionnarie in supplementary material refered in lines 186-187

Line 264: Please clarify if the monetary unitary is dollars or another currency.

The currency was dollars, we included the currency.

Line 345: Add the units for each axis.

Added

General comments

It could be useful to include a list of cut-off points for each biochemical definition.

Use a point to indicate decimal values.

Thank you for the observation. The table detailing the cut-off points for the variables is already included as supplementary material and refered in the main text in lines 239-240. Additionally, the commas have been as suggested.

---

## [Decision Letter · Decision Letter 2]

27 Aug 2025

The Burden of Non-Communicable Disease Risk Factors in a Low-Income Population: Findings from a Cross-Sectional Study Highlighting the Prevalence of Obesity, Hypertension, and Metabolic Disorders in the south of Quito, Ecuador.

PONE-D-24-46891R2

Dear Dr. Morales-Garzón,

We’re pleased to inform you that your manuscript has been judged scientifically suitable for publication and will be formally accepted for publication once it meets all outstanding technical requirements.

Kind regards,

José M. Alvarez-Suarez

Academic Editor

PLOS ONE

Additional Editor Comments (optional):

Reviewers' comments:

Reviewer's Responses to Questions

**Comments to the Author**

1. If the authors have adequately addressed your comments raised in a previous round of review and you feel that this manuscript is now acceptable for publication, you may indicate that here to bypass the “Comments to the Author” section, enter your conflict of interest statement in the “Confidential to Editor” section, and submit your "Accept" recommendation.

Reviewer #1: All comments have been addressed

2. Is the manuscript technically sound, and do the data support the conclusions?

Reviewer #1: Yes

3. Has the statistical analysis been performed appropriately and rigorously? 

Reviewer #1: Yes

4. Have the authors made all data underlying the findings in their manuscript fully available?

Reviewer #1: Yes

5. Is the manuscript presented in an intelligible fashion and written in standard English?

Reviewer #1: Yes

6. Review Comments to the Author

Reviewer #1: The authors have solved all the comments made in the previous version. I dont have specific or additional comments.

7. PLOS authors have the option to publish the peer review history of their article (what does this mean? ). If published, this will include your full peer review and any attached files.

**Do you want your identity to be public for this peer review?** For information about this choice, including consent withdrawal, please see our Privacy Policy .

Reviewer #1: **Yes: ** Sandra Cortés

---

## [Editor Report · Acceptance letter]

PONE-D-24-46891R2

PLOS ONE

Dear Dr. Morales-Garzón,

I'm pleased to inform you that your manuscript has been deemed suitable for publication in PLOS ONE. Congratulations! Your manuscript is now being handed over to our production team.

Kind regards,

on behalf of

Professor José M. Alvarez-Suarez

Academic Editor

PLOS ONE